# Lifesaving Treatments for the Tiniest Patients—A Narrative Description of Old and New Minimally Invasive Approaches in the Arena of Fetal Surgery

**DOI:** 10.3390/children10010067

**Published:** 2022-12-28

**Authors:** Thomas Kohl

**Affiliations:** German Center for Fetal Surgery & Minimally-Invasive Therapy (DZFT), Mannheim University Hospital (UMM), Theodor-Kutzer-Ufer 1-3, 68167 Mannheim, Germany; thomas.kohl@umm.de; Tel.: +49-175-597-1213; Fax: +49-621-383-5979

**Keywords:** fetal surgery, fetal intervention, minimally invasive therapy, diaphragmatic hernia, CPAM, spina bifida, hydrothorax, LUTO, hyperoxygenation

## Abstract

Fetal surgery has become a lifesaving reality for hundreds of fetuses each year. The development of a formidable spectrum of safe and effective minimally invasive techniques for fetal interventions since the early 1990s until today has led to an increasing acceptance of novel procedures by both patients and health care providers. From his vast personal experience of more than 20 years as one of the pioneers at the forefront of clinical minimally invasive fetal surgery, the author describes and comments on old and new minimally invasive approaches, highlighting their lifesaving or quality-of-life-improving potential. He provides easy-to-use practical information on how to perform partial amniotic carbon dioxide insufflation (PACI), how to assess lung function in fetuses with pulmonary hypoplasia, how to deal with giant CPAMS, how to insert shunts into fetuses with LUTO and hydrothorax when conventional devices are not available, and how to resuscitate a fetus during fetal cardiac intervention. Furthermore, the author proposes a curriculum for future fetal surgeons, solicits for the centralization of patients, for adequate maternal counseling, for adequate pain management and adequate hygienic conditions during interventions, and last but not least for starting the process of academic recognition of the matured field as an independent specialty. These steps will allow more affected expectant women and their unborn children to gain access to modern minimally invasive fetal surgery and therapy. The opportunity to treat more patients at dedicated centers will also result in more opportunities for the research of rare diseases and conditions, promising even better pre- and postnatal care in the future.

## 1. Biography

Thomas Kohl is a pediatrician with 20 years of clinical experience in minimally invasive fetal surgery. He developed, co-developed, and clinically introduced a large spectrum of fetal interventions. He pioneered the technique of fetoscopic closure of spina bifida, as well as the technique of fully percutaneous partial amniotic carbon dioxide insufflation (PACI). He co-developed fetoscopic tracheal balloon occlusion for the treatment of severe pulmonary hypoplasia from diaphragmatic hernia, anhydramnios, hydrothorax, and CPAM. He performed the first successful fetoscopic procedures in fetuses with laryngeal atresia and introduced the techniques of fetal intraamniotic ultrasound, fetal transesophageal echocardiography, and fetal transesophageal electrocardiography. Furthermore, Kohl has developed new, minimally invasive techniques for early second trimester shunt insertion in fetuses with LUTO and for fetuses with hydrothorax. He considers the treatment of hypoplastic cardiovascular structures in human fetuses by chronic intermittent materno–fetal hyperoxygenation his most important contribution. He is the founder of the German Center for Fetal Surgery & Minimally-Invasive Therapy (DZFT).

## 2. Introduction

Following its introduction in the 1980s by Michael Harrison, Mickey Golbus, and Roy Filly, fetal surgery has come a long way. In particular, the development of minimally invasive surgical techniques since the early 1990s has led to an increasing acceptance of novel procedures. As a result, thousands of patients worldwide have been saved by prenatal interventions.

The founding years of fetal surgery were met with numerous challenges and unknowns regarding maternal and fetal welfare [1]. Maternal laparotomy and hysterotomy defined the approach of “open” fetal surgery. Both were required to get access to the fetal patient for draining an obstructed bladder or diaphragmatic hernia repair [2]. As these procedures were pioneered solely in animal models, the likelihood of a human fetus with these life-threatening conditions to benefit from such novel experimental treatment was essentially unknown. Additionally, anesthetic and tocolytic management of prenatal surgical candidates—in the absence of experience and well-established safety parameters—posed significant risks to both the mother and fetus. Evidence derived from animal studies alone could not fully address the technical feasibility and therapeutic potential of these procedures on a delicate human fetus. Despite the originally intended scope of open fetal surgery, the only remaining indications for this invasive surgical approach are spina bifida repair, sacrococcygeal teratomas, and lung malformations [2], although in recent years all of these malformations have also become amenable to minimally invasive fetal interventions.

Manuscripts and chapters have been published to address the technical, ethical, and legal ramifications of the new field and to provide a framework for its conduct [1,2,3]. Regarding animal experimentation, Harrison and colleagues acknowledged the need for each surgically induced model of a fetal malformation to resemble the known human pathology as closely as possible. Furthermore, prenatal intervention on that model by a second procedure later in gestation was expected to show that the pathologic development would be arrested, improved, or—at best—even reversed. In other conditions, such as twin-to-twin transfusion syndrome (TTTS) where animal models were not available, at least the technical steps could be rehearsed before clinical introduction [3].

As safety was paramount, fierce discussions between opponents and proponents were held regarding the maternal and fetal risks and ethical consequences of the fetus becoming a patient [4,5]. Some of these altercations have survived over the decades, driven by cemented opinions, a fear of change, envy, animosity, and other undesirable human traits. The good news for today’s fetal surgery patients is that right now, the generation of the fiercest opponents of fetal therapy, who disregarded or hindered much of its progresses over the decades, is retiring.

Fetal surgery has never been for the fainthearted. As in other fields of surgery, the initial setbacks and outcomes while pioneering surgical procedures would have scared most individuals with a normal decorum away. The field “would have to draw the most aggressive, most daring of doctors; those who were ready to try almost anything to save a fetus” [5]. Early fetal surgery was about doctors daring to perform procedures that at the time involved significant risks for the mothers, too. A brazen example of the fierce competition for spectacular firsts is provided in the episode “Saving Life Before Birth” of the 2020 Netflix docuseries “*The Surgeon’s Cut*”, when the clinical introduction of the first fetoscopic laser ablation for the treatment of twin-to-twin transfusion syndrome is falsely claimed by the protagonist and his colleague. This feat has been attributed to Julian De Lia as the first author on the first publications about technical animal studies and the early clinical experience with treating this condition [6,7].

Today, apart from a small group of truly highly trained and skilled practitioners, the field of fetal intervention attracts mainly Ob/Gyn-trained diagnosticians working in private practice, general hospitals, or university hospitals. Unfortunately, many lack the resources and the expertise at their home institutions to ascertain the training and mentorship necessary for fetal surgery. Aside from the insufficient pedagogy, most clinicians who show interest still lack both the patients and the patience for learning these interventions thoroughly and safely. In fact, it is hard to think of any other discipline in medicine, where patients at the genesis of morbidity or mortality could be treated—and cured—based on medical licensure alone.

Since most fetal interventions are performed on fetuses with critical pathologies, treatment failures are often easily dismissed by the preconception that a gravely ill fetus would have died anyway. On the other hand, any avoidable failures or serious complications are used to validate the prejudices held against the practice of fetal surgery. Thus, the fate of each delicate unborn patient is held in the balance of continued fetal intervention and advocacy, with the responsibility of informed obstetricians and fetal medicine specialists to provide unbiased counseling on its benefits and risks.

Unfortunately, fetal surgery has not yet become a board-certified subspecialty. In the beginning, this fact was due to low patient numbers, its experimental character, and the ethical and medicolegal concerns at hand. Despite 40 years of progress—where in countries such as ours today (Germany), several hundred fetuses are treated each year—this reality is mainly of political motivation. There are numerous academic gynecologists who hold on to the fears of giving valuable ground and patients away to newly specialized practitioners. There are also medical faculties who remain in the shadows, uninformed and unaware of the enormous potential of this new field. Intense efforts are necessary to ensure that the field of fetal surgery will receive academic resources, as well as adequate support and respect of the medical community, as this will be lifesaving for future fetuses.

Yet, in contrast to firmly held beliefs, training in pediatric surgery, neonatology, pediatric cardiology, or pediatric nephrology alone is not sufficient for best serving the needs of unborn patients; fetuses and premature infants or neonates—and their physiologic habitats—are all too different. Furthermore, training in pediatric subspecialties alone does not aid in addressing the periinterventional needs of the mother—the innocent bystander—whose safety and integrity are paramount concerns to the practice of fetal surgery. So—much to the surprise of the practitioners of open fetal surgery—it was the pioneering obstetrician–gynecologist Ruben Quintero who envisioned, developed, and clinically introduced the first spectrum of truly minimally invasive, percutaneous fetoscopic approaches [8].

### 2.1. What It “Should” Take to Become a Fetal Surgeon

Fetal surgery is a complex field and benefits from the knowledge and practical experience of many other disciplines: The maternal management of fetal surgery patients can mostly be learned through a comprehensive residency program in obstetrics and gynecology at a center performing fetal surgery. Ultrasound skills must be learned during special maternal fetal medicine fellowships, supplemented by special ultrasound courses. Then, a basic theoretical and practical skill set would have to be learned by dedicated residents that reproducibly and safely permit them to perform a range of interventions. I highly recommend learning the interventions and the handling of interventional materials in inanimate models, simulators, or postmortem interventions before treating unborn babies. In the end, they should be able to place the needle tip during an ultrasound-guided puncture with mm-precision wherever required, and even when imaging quality is poor. Such a rigorous training approach should become mandatory due to the very limited margins of error for most fetal interventions; the slightest mishap may render a procedure unsuccessful, lethal, or impact negatively on its therapeutic outcome.

In order to manage any upcoming periinterventional issues, this journey requires several years of training in embryology, fetal, and neonatal pathophysiology, obstetrics, neonatology, pediatric surgery, fetal cardiology, pediatric cardiac and non-cardiac intensive care, as well as prenatal medicine and genetics. Last but not least, acquiring sufficient acumen for academic research and publication is also necessary as the development and establishment of new techniques requires their analysis and dissemination.

This schedule provides a rough framework for becoming acquainted with the field and developing a basic competency with the procedural tasks. Mastery in fetal surgery, however—as in any other field—can only be achieved by succumbing to the ritual of life-long learning. This would include the deliberate training of rare procedures and complication management, constant striving to perform the best possible interventions, and an honest acknowledgment of and humbly learning from failures. In this vein, the ground-breaking studies of Ericsson, Krampe, and Tesch-Romer and the books of Colvin and Leonard provide important insights into how to begin and stay on this rewarding yet arduous path [9,10,11]. Furthermore, I would recommend playing piano. This recommendation may surprise you at first. Yet this instrument is an optimum daily training tool for achieving tremendous amounts of dexterity, coordination, and freedom of movement. These skills are of major advantage for any endoscopic surgeon or interventionist who needs to steer his instruments with utmost precision.

### 2.2. Materno–Fetal Pain Management and the Safety of Fetal Surgery

Maternal safety should always be the primary concern of any fetal intervention. Surprisingly, even today, many fetal interventions are performed and taught under questionable hygienic circumstances. A stunning example of the director of a leading international teaching organization for fetal therapy disregarding all principles of antisepsis is available on the Internet: https://www.youtube.com/watch?v=ojy8qUN1CKw (accessed on 26 December 2022).

Procedures performed without adequate antisepsis precautions, materno–fetal analgesia, or sedation not only subject mothers and fetuses to avoidable infections as well as stress and pain, but they may also interfere with the technical success and overall outcome of the procedure. Once again, these problems aid in keeping the prejudices held against the practice of fetal surgery alive.

At our center, effective materno–fetal pain management is one of the most important management pillars that allows for the most difficult, percutaneous, minimally invasive ultrasound-guided or fetoscopic procedures to be performed—in both hemodynamically stable and compromised fetuses [12]. Earlier assumptions that minimally invasive fetoscopic surgery would require a high dosed regime of materno–fetal anesthetics for uterine relaxation and the prevention of preterm contractions similar to the open operative approach proved to be false. On the contrary, we learned that for both intravenous and inhalation anesthetics, dosages at the lower end of the spectrum do indeed suffice [13]. As a result, severe maternal hypotension from deep anesthesia—previously requiring maternal administration of high volumes of crystalline fluids and high dosages of catecholamines—is no longer seen during fetoscopic surgery. Hence, maternal pulmonary edema, fetal intraoperative demise, or fetal brain injury are not on our list of severe adverse events from minimally invasive fetal surgery. In contrast, these complications are still observed following the invasive open operative approach [14].

### 2.3. Partial Amniotic Carbon Dioxide Insufflation (PACI)

After years of animal studies in sheep, I clinically introduced a fully percutaneous technique for partial amniotic carbon dioxide insufflation (PACI) in humans 20 years ago [15,16,17,18,19,20,21]. PACI has been invaluable in improving the visualization of both the fetus and intraamniotic contents during complex fetoscopic procedures. PACI was instrumental in achieving the minimally invasive fetoscopic patch closure of fetuses with spina bifida. The technique also greatly facilitates removal of amniotic bands, fetal posturing, resection of tumors, laser ablation of twin-to-twin transfusion-syndrome in the presence of an anterior placenta, and the performance of some cardiac interventions (Figure 1).

In the following section, I want to provide a how-to-manual on how to safely perform PACI in order to further promote the dissemination of this lifesaving technique:When planning to perform PACI during general materno–fetal anesthesia, ventilate maternal CO_2_ levels within the physiologic range of 30–35 mm Hg. Use muscle relaxants during longer procedures in order to keep the insufflation pressures as low as possible. Maintain maternal systolic blood pressures around 120 mm Hg.When employing a fully percutaneous setup, place all trocars that you want to use before insufflation. This is particularly important when the placenta is situated anteriorly.If the workspace is too narrow, perform an amnioinfusion with crystalloid solution before insufflation. Limit the infusion to no more than 400–600 mL, which is an amount that is usually well tolerated and does not result in excessive overdistension.In the case of more severe polyhydramnios, perform an amniodrainage until the uterine tone begins to soften. This maneuver is important to avoid excessively high insufflation pressures.Before insufflation, once again check the feto–placental blood flow. If you encounter absent or reverse end diastolic blood flows within the umbilical artery (which is rare; estimated 1–2%), do not begin with the insufflation. In case you cannot obtain a normalization of the Doppler flow profile by normalizing the maternal blood pressure or utero–placental blood flow by maneuvers such as corrections of maternal lie, volume administration, catecholamines, or amniodrainage, abort the procedure. You may try another day with a different anesthetic protocol of modified dosages. If you have a normal fetoplacental Doppler flow signal, you can begin with the insufflation.Begin with insufflator settings of 8 mm Hg insufflation pressure and a flow rate of 4–6 L/min. Press the insufflator start button and fill the insufflation tubing with carbon dioxide. In case you insufflate via several trocars, connect the various tubes to all but one of the trocar locks and keep their valves closed. Now open the valve of the last trocar and bring the insufflation tubing and the lock in close contact. Now reset the flow meter to “0” and immediately connect the tubing with the open trocar lock. Have an assistant hold up the tubings such that the amniotic fluid can not escape into the main insufflation tube or the insufflator.Now increase the insufflation pressure of the insufflator by increments of 2 mm Hg until the flow meter measures the release of carbon dioxide into the amniotic cavity. This pressure is what I define as the “opening pressure”. Once you have reached the opening pressure, you insert the fetoscope and inspect the fetus, placenta, and amniotic cavity. Insufflation pressures can range between 8 mm Hg (in multipara) up to 30 mm Hg or more (nullipara, polyhydramnios, stressed patients). The mean insufflation pressure is about 15 mm Hg.If you operate via only one trocar, insert a suction rod and remove a sufficient amount of amniotic fluid such that you can perform the procedure. If you operate via several trocars, you can monitor the removal of amniotic fluid with the suction rod by fetoscopy.

Caveats:The workspace must be checked by paying attention with the fetoscope every 5–10 min.Overinsufflation must be avoided by all means in order to avoid placental abruption and prevent chorioamniotic separation.The lower the opening pressure, the higher the risk for overinsufflation.Very high opening/insufflation pressures beyond 20 mm Hg are well tolerated. Yet over the course of the procedure, uterine tone often decreases, demanding a substantial downregulation of the insufflation pressure.Over the course of the insufflation, the workspace may become smaller for three reasons: first, from increases in uterine tonus (deepen anesthesia), second, from increases in maternal abdominal muscle tonus (provide additional muscle relaxants), and third, from continuous leakage of small amounts of amniotic insufflation gas into the maternal abdominal cavity resulting in an equalization of pressures (evacuate gas by placing an additional hemodynamic sheath or 14-gauge indwelling catheter into the maternal peritoneal cavity).Special care must be taken when a trocar dislodges or during extensive trocar excursions in the presences of high insufflation pressures beyond 20 mm Hg! In this situation, a sudden gas leakage along the trocar shaft into the maternal abdominal cavity may occur. The sudden steep increase of intraabdominal pressure may acutely impair infracardiac venous return and result in a sharp drop of maternal cardiac output. This potentially dangerous situation is immediately corrected by evacuation of the insufflation gas by suction or opening of all trocar valves.The most feared complications are maternal gas embolism or fetal demise from acidosis. From September 2002 to December 2022, I have used PACI more than 300 times and have never observed either event. Over the past decade, PACI has been adopted and its safety has been confirmed by a growing number of international fetal centers [22,23,24].In my opinion, warming and moistening the insufflation gas is not a definitive prerequisite, as the insufflation gas volumes used are small and mostly stagnant such that the intraamniotic conditions swiftly serve the same purpose.

### 2.4. Fetoscopic Tracheal Balloon Occlusion in Fetuses with Diaphragmatic Hernia

The technique of fetal tracheal balloon occlusion (Figure 2) was developed in order to salvage fetuses with life-threatening pulmonary hypoplasia from left and right diaphragmatic hernias [25,26,27,28,29]. The idea to use this method originated from experimental animal studies that showed that preventing the efflux of lung fluid during fetal life by tracheal ligation results in marked lung distension and growth [28] and from studying “CHAOS” (congenital high airway obstruction syndrome), an experiment of nature in which the same effect can be observed in human fetuses with laryngeal or tracheal atresia.

A major problem was to find a suitable material for fetal tracheal occlusion, so different means were tested. Michael Harrison and his team tested external constriction by hemostatic clips [25], Ruben Quintero employed an intratracheal plug [26], and Eurofetus investigators and our group worked with intratracheally detachable balloons borrowed from neuroradiology [29,30]. The early operative approaches were invasive hybrid and open fetal surgical procedures. Again, it was Ruben Quintero who introduced the first truly minimally invasive fully percutaneous approach in a human case [26].

Already in 2006, our study group designed and registered two randomized trials to test our fully percutaneous minimally invasive FETO approach with a detachable latex balloon in severe and moderate cases (ClinicalTrials.gov Identifier: NCT00373763 and NCT00373438). Yet, our efforts failed, as eligible expecting mothers in our country refused for their unborn babies to be randomized to the fates of life or death. These patients found the backdoors to competing European centers wide open.

As a result, FETO was applied for even more years as an experimental and potentially lifesaving treatment approach. Then, Rodrigo Ruano and colleagues were the first to complete a randomized trial in 2012 that showed an impressive survival advantage for fetuses with severe diaphragmatic hernia and intrathoracic liver herniation (liver-up) when prenatally treated by FETO (50%) compared to those prenatally untreated (5%) [31]. Yet doubts about its validity were raised because two different calculations of fetal lung size were used for inclusion.

In two larger, more recent trials, Deprest and co-investigators of the Eurofetus consortium confirmed Ruano’s findings that FETO results in statistically significant improvement of survival in fetuses with severe diaphragmatic hernias (o/e-lung-to-head-ratio < 25%) [32,33]. However, by selecting fetuses for FETO “irrespective of liver position”, the investigators of the Eurofetus trial departed from all previous studies and employed a study design with a bias toward better outcomes.

In this light, the quite low rates of fetal survival in both their treatment and control groups are a disappointment and result from the chosen international, multicenter trial design. Extracting the value of FETO from up to 10 prenatal and 26 neonatal treatment centers of up to 12 European countries can only result in average outcomes. As a consequence, the trial results can neither reflect the true potential of prenatal therapy nor that of postnatal therapy in severe and moderate cases of diaphragmatic hernia.

The poor results of the TOTAL trial not only discourage new expectant mothers and their partners. The low survival rates of both FETO trials also discouraged prenatal medicine specialists and neonatologists at our center, which treats as many as 70 neonates each year with diaphragmatic hernias.

This lack of enthusiasm is based on the fact that in dedicated high-volume-ECMO-centers, the survival rates for fetuses with moderate to severe right and left diaphragmatic hernias are 10–20% higher without FETO than in the trial fetuses with FETO. Overall, a 70% survival rate of fetuses with moderate to severe left diaphragmatic hernias and an 80% survival rate of fetuses with moderate to severe right diaphragmatic hernias can be achieved [34,35].

Therefore, whenever and wherever possible, fetuses with moderate to severe diaphragmatic hernias should preferentially be referred to dedicated ECMO centers. At least in Europe, this strategy is possible, given the general availability to issue a “certificate concerning the retention of the right to sickness or maternity benefits currently being provided” (S2/E-112) for cost coverage.

As an alternative, in countries with less access to the best possible care and fewer financial options, the chances to improve postnatal outcomes might be estimated based on the availability of local resources and from the known outcomes of previous cases. If at a center the chances of survival for mild to moderate cases are low and referral to a dedicated center is not possible, the selection threshold for offering FETO as a means to alleviate the postnatal treatment course may be lowered substantially. And, even in severe cases, FETO has shown a survival benefit, when postnatal management is known to be suboptimal [36].

Yet FETO comes with a specific risk of asphyxiation, which endangers even those fetuses who at first show its desired therapeutic effects. In some cases, it has been in vain to remove the tracheal balloon at delivery. This has happened most often when pregnant women carrying fetuses with an occluded trachea had been sent home after the procedure and rapid delivery had to be performed in a different hospital unfamiliar or inexperienced with removal or destruction of the balloon [37]. In order to prevent such catastrophic events from happening, our policy has always been to keep FETO patients in our hospital and provide all of the necessary resources for balloon removal in the delivery unit.

A promising alternative that should improve the safety and reliability of balloon removal is the “smart” tracheal balloon developed by Eurofetus investigators. The new technology allows for the balloon to be deflated within the magnetic field of an MRI scanner [38]. It must now undergo rigorous safety and efficacy tests before it hopefully might become commercially available in the coming years.

Unfortunately, by the end of July 2022, the producer of the currently employed latex balloon for FETO will not recertify its product. This crisis has prompted a stampede by European FETO centers to secure as many devices as possible. This event highlights another typical problem in the field of fetal surgery, where the majority of surgical devices and interventional materials are sourced from other pre-existing procedural sets instead of being designed specifically for use in fetal intervention. Treating rare diseases very rarely encourages medical device companies to design, manufacture, and certify adequate products. As a result, most devices used during minimally invasive fetal interventions are used off-label.

### 2.5. Additional Hemodynamic Assessment of Fetuses with Diaphragmatic Hernia

In most centers, the severity of fetal diaphragmatic hernias has been determined by assessment of their liver-position (“up” or “down”), the observed-to-expected lung-to-head ratio (o/e-LHR) measured during ultrasound examination, or the observed-to-expected lung volume assessed by MRI [39,40]. Although these anatomic parameters have been useful estimates for quantifying the chances of postnatal survival, they fail to explain why some fetuses that were expected to have a good prognosis fare worse during postnatal treatment or vice versa.

In order to overcome this limitation, additional hemodynamic studies of fetuses with diaphragmatic hernia—with a particular focus on fetal left–right-heart-symmetry—can be performed, such as assessment of ductus venosus streaming and lung blood flow monitoring [41,42,43]. Using these parameters, I have observed that most fetuses with liver-down left diaphragmatic hernia exhibit normal streaming of ductus venosus blood into the left side of the heart of the fetus. As a result, both sides of the heart are of equal size, pulmonary blood flow signals are easily visualized, and pulmonary vasodilation in response to diagnostic maternal hyperoxygenation can be produced (Figure 3). Taken together, these findings point to a milder postnatal treatment course and herald a better prognosis.

In contrast, most fetuses with liver-up left diaphragmatic hernia exhibit preferential streaming of ductus venosus blood into the right side of the heart of the fetus. As a result, there is less preload to the left side of the heart, which in turn becomes hypoplastic. Probably as a result of the oxygen-enriched ductus venous blood reaching the lungs, morphologic changes contributing to pulmonary hypertension can develop. During Doppler ultrasound studies, pulmonary blood flow signals are less pronounced and pulmonary vasodilation in response to materno–fetal hyperoxygenation is often markedly attenuated (Figure 3).

Taken together, these findings point to a more complicated postnatal treatment course, requiring ECMO, and to a more guarded prognosis [44]. Only 10% of fetuses with liver-up left diaphragmatic hernia present with normal ductus venosus streaming, which may give them a better prognosis.

Importantly the study of the ductus venosus flow direction may aid in identifying the about 10–20% of fetuses with liver-down left diaphragmatic hernia that are at risk of a more severe postnatal course and the need for ECMO, as this subgroup also exhibits preferential ductus venosus streaming to the right side of the heart, left heart hypoplasia, and impaired lung blood flow.

In contrast, fetuses with right diaphragmatic hernia—despite liver herniation—present with a more normal infracardiac spatial arrangement of venous vessels, and hence, normal distribution of ductus venosus blood within the fetal heart [43]. Therefore, as a rule, in these fetuses, *neither isolated left, nor isolated right heart hypoplasia are observed* (I have encountered only one exception) (Figure 3). Nevertheless, there may be *symmetrical* underdevelopment of both, left and right cardiovascular structures, from compression by the herniated organs. Yet even in severe cases, a close to normal degree of blood flow can be observed within the left lung (Figure 3). In ECMO centers such as ours, this hemodynamic advantage seems to contribute to survival rates of more than 80% for this condition [34].

### 2.6. Fetoscopic Tracheal Balloon Occlusion (FETO) in Fetuses at Risk for Severe Pulmonary Hypoplasia from Other Conditions

In my experience, FETO is also effective in fetuses at risk for the development of severe pulmonary hypoplasia from long-standing anhydramnios from the very early preterm rupture of membranes [45,46], from renal agenesis, from poor lung distension after successful drainage of bilateral hydrothorax, or from compression by giant, non- or poorly regressing congenital pulmonary airway malformations (CPAM).

When other treatment options fail in these conditions or gestation is too advanced to permit enough time for the sufficient catch-up growth of the fetal lungs, fetoscopic tracheal balloon occlusion (FETO) followed by postnatal treatment at a dedicated neonatal lung treatment center may increase the chances of postnatal survival.

Particularly in fetuses with anhydramnios from preterm rupture of membranes prior to 20 weeks of gestation—the speed of pulmonary catch-up growth and the improvement in lung perfusion observed during FETO by far surpasses the ones observed in fetuses with diaphragmatic hernias [44] (Figure 4). This advantage can be explained by the presence of principally intact and far more developed lungs on either side of the chest at the time.

### 2.7. CPAM

Congenital pulmonary airway malformations (CPAM) constitute a spectrum of lung malformations that only in more severe cases require fetal intervention [47]. Given the high likelihood of spontaneous regression over the final third of gestation, there is a consensus among fetal interventionists to treat only those malformations that, due to their size or vascular steal, are accompanied by fetal hydropic changes and may jeopardize fetal survival.

Various prenatal treatment strategies have been employed for rescuing the affected fetuses. Among these are thoraco-amniotic shunt placement into large cystic lesions, embolization, radiofrequency ablation or laser ablation of feeding vessels in case of lung sequestrations and percutaneous sclerotherapy [48,49,50,51]. In addition, steroid therapy has been used in hydropic fetuses with large microcystic lesions [52].

Any of these approaches aims to achieve a substantial volume reduction of the malformation. The resulting drop in central venous pressures, improved cardiac filling and output aid in the quick resolution of hydropic changes. Yet, in many centers around the world, the treatment options described above may not be available. In other instances, they may be in vain or result in untoward complications [52,53]. Thus, new strategies were needed to address these setbacks and to develop a straightforward and universally available approach.

Inspired by the simple and highly effective treatment of alcohol ablation of a Bartholin cyst abscess [54], I began treating very large cystic CPAM accompanied by fetal hydrops by alcohol ablation. The rationale behind this approach is that the alcohol destroys the inner fluid-producing cell layer of the cysts, providing a means of shrinking the lesion with immediate effect.

In brief, materno–fetal anesthesia is initiated to guarantee both maternal and fetal immobilization. After surgical cleansing and draping of the maternal abdomen, a 2–3 mm skin incision is made, promptly followed by the insertion of an 18- to 20-gauge trocar needle directed into the largest proximal cyst. Subsequently, the fluid is aspirated from the cyst and its communicating cystic parts until the punctured structure collapses almost entirely. This step is followed by instillation of 5–10 mL of 95% ethanol (BRAUN, Melsungen, Germany) into the cyst system. After 5 min of ablation, the ethanol is withdrawn, and the needle is removed (Figure 5).

Importantly, this approach may also be used to shrink large microcystic lesions which usually are the hardest to treat. In these cases, it is important to consider that the instilled amount of ethanol cannot be withdrawn. Therefore, only very small volumes of 0.1–0.2 mL ethanol have to be used. These must be instilled at multiple locations within the malformation, minimizing the risk of ethanol spillage into the pleural cavity, which could disturb the normal development of the fetal chest wall. This has been the only complication of ethanol ablation we have observed in one of our survivors. Other groups with experience of sclerotherapy with 5% ethanolamine oleate report that its intravascular injection should be avoided because it may result in cardiac necrosis [55,56].

Following long-standing compression of the ipsilateral and contralateral lung tissue by very large CPAMs—even after technically successful interventions—some lungs do not distend and do not exhibit improved perfusion and are considered too small or too dysfunctional for postnatal survival. As mentioned previously, fetoscopic tracheal balloon occlusion (FETO) may serve as the last resort to achieve lifesaving lung catch-up growth in these cases (Figure 6).

### 2.8. How to Detect Fetuses with Life-Threatening Pulmonary Hypoplasia

There is no simple ultrasound scoring system in the literature that predicts postnatal lung function in fetuses with pulmonary hypoplasia from preterm rupture of membranes or other origins of anhydramnios [57,58]. Therefore, many neonatologists and expectant parents of babies at risk have surrendered almost entirely to a wait-and-see-approach. As several prenatal therapeutic options are available that may prevent the development of lethal pulmonary hypoplasia, it is of paramount importance to abandon this approach and proactively define low- and high-risk groups such that the latter ones can be scheduled for intervention.

As early as 15 years ago, I began developing such a scoring system for myself. Over the years it has been tremendously helpful for determining the need for fetal intervention and—in case of FETO or serial amnioinfusions—the optimum time to initiate these treatments, as well as the timing of any subsequent procedures and even the timing of delivery. Having been held up by a never-ending flow of other obligations, I was never able to systematically study and publish it. Therefore, I want to present the essence of this precious beacon in this manuscript such that its use and value can be studied and tested by other clinicians.

How does it work? The fetus is studied by a 2D-ultrasound in horizontal and vertical cross-sections through the chest. The heart–lung area relationship and the shape of the thorax (normal vs. bell-shaped) are observed and the presence or absence of pseudocardiomegaly is noted (Figure 4). Then, the height of the lower borders of the heart and lungs (heart–lung base) are determined in para-sagittal sections through the chest. Finally, a 3 × 3 cm lung section (closest to the transducer) from the chest wall to the posterior border of the heart is interrogated by color Doppler ultrasound at a Nyquist limit of about 15 cm/s, employing the highest degree of amplification before color flickering occurs. The presence or absence of color signals within the lung area is noted. In fetuses beyond 26 weeks of gestation, diagnostic materno–fetal hyperoxygenation with a face mask (flow 6 L/min) is used to assess the presence or lack of pulmonary vasodilation in this area using the same ultrasound settings.

How to interpret the study?

-Fetuses with normal or near normal heart–lung area relationships (i.e., a normal four-chamber view), a normal chest shape, a heart–lung base below the 5th rib (i.e., normal/near normal cardiac four-chamber views), and well demonstrable color flow signals are at low risk.-Fetuses with the beginnings of pseudocardiomegaly, a heart–lung base around the 4th rib, demonstrable lung areas within either chest cavity, and demonstrable color flow signals are at moderate risk. For their treatment, oxygen, NO, and surfactant as well as chest tubes should be readily available in the resuscitation suite.-Fetuses with marked pseudocardiomegaly, a heart–lung base around or higher than the 4th rib, a bell-shaped thorax, only small lung areas within either chest cavity (Figure 4), and barely or no demonstrable color flow signals during hyperoxygenation are those with the highest risk of dying. The need for immediate and intense resuscitation efforts can be expected, and the entire available arsenal must be employed. If the clinical status of the mother allows it, it is this latter group that I have chosen for FETO.

To confirm the ultrasound findings from before and after these interventions, we also use fetal MRI studies to assess lung volumes. Here, the hypointense, dark lung tissue during T2-imaging appears to be the most important predictor for a complicated postnatal treatment course. The hypointensity seen on MRI corresponds well with the poor pulmonary blood flow signals that are found in severely affected fetuses during color Doppler interrogation. After successful interventions such as FETO or serial amnioinfusions, the MRI intensity and lung blood flow improve, heralding lungs that can be postnatally treated with good prognosis.

### 2.9. Improving the Outcome of Fetuses with Lower Urinary Tract Obstruction (LUTO)

Over the past decade, a major project has been working to improve the chances of fetuses with severe lower urinary tract obstructions (LUTO) to survive with better preserved renal and pulmonary functions. Following the publication of the aborted randomized PLUTO trial, the poor outcome of the prenatally treated group prompted many groups to abandon vesico-amniotic shunting (VAS) for good [59]. Although more fetuses survived in the VAS group, only about 20% demonstrated normal renal function after delivery, which was similar to the prenatally treated and untreated groups.

Most centers have performed VAS from 18 weeks of gestation onwards based on previous personal or published experience that earlier shunting with the often used double-pigtail catheters is unsafe. The cause is two-fold; until about 15 weeks of gestation, physiologic separation of the chorioamniotic membrane is present and makes early interventions more prone to the early preterm rupture of membranes. This anatomical hurdle is worsened by the fact that placement of a double-pigtail catheter commonly requires additional amnioinfusion, along with more extensive needle excursions, in order to position the external pigtail end within the amniotic cavity.

Nevertheless, based on clinical observations and seminal studies by Michael Harrison and his colleagues in fetal sheep in the early 80 s [60,61], mimicking the natural history of severe obstructions, I was convinced that late first or early second trimester treatment would provide a better chance of preserving kidney and lung function than what had been achieved by later treatment beyond 16 weeks of gestation [62].

In order to achieve early second trimester shunting as safely as possible, I developed a novel shunt technique for megacystis in 2012. The hallmark of this technique is the use of a single-pigtail catheter that permits vesico-amniotic shunt insertion without any manipulation of fetal position, little strain on the uterine wall and membranes, and no need for amniotic fluid augmentation. The procedure is performed under materno–fetal anesthesia within usually less than five minutes. All of the materials are widely available in hospitals with adult and pediatric intensive care.

In short, following a small maternal skin incision, an 18-gauge needle is advanced into the fetal bladder under ultrasound guidance, preferably via the lower half of the fetal abdomen below the umbilicus (Figure 7). Then, a stiff 0.035′′ guide wire is inserted through the needle shaft, followed by removal of the needle. The wire serves as a rail for the insertion of an 8-F-catheter sheath. This sheath is then used for placing an adequately trimmed single pigtail catheter into the bladder. Then, the catheter sheath is withdrawn without any further manipulation and the incision is closed with a single suture. As soon as the fetus awakens from anesthesia and begins moving again, it pulls the distal catheter end into the amniotic cavity, thereby completing the procedure. Most often, this step does not take longer than a day. Using this literally straightforward approach, amnioinfusion and extensive device manipulations are unnecessary, thus decreasing the risk of the preterm rupture of membranes and enabling early interventions.

This technique was rivaled by the introduction of a far more expensive stent catheter by the company SOMATEX (Berlin, Germany) which also permits more successful and safer early second trimester shunting in fetuses with megacystis [63]. Unfortunately, because of its design (parasols at both ends) and its limited length, dislodgment or trapping can be observed, such that the need for amnioinfusion and repeated placements accompany its use. As an alternative, the Q-shunt that cannot dislodge or be pulled out by the fetus was introduced by Quintero and colleagues [64]. Yet another recent approach that foregoes shunt placement uses a coronary angioplasty catheter via a guide-wire approach for prenatal dilation of posterior urethral valves [65].

Apart from improving the technical aspects of the earlier intervention, I intentionally forwent any pre-interventional diagnostic procedures. This includes chorionic villus sampling, amniocentesis, and serial urinary taps, in order to avoid detrimental treatment delays. Preferably, amniotic fluid is withdrawn for genetic testing at the time of VAS. Through this strategy, the time of severe urinary obstruction is kept as short as possible, which in our series has resulted in far better renal preservation in survivors than what had been described in the literature for decades: normal values for renal parameters were found in about 80% of survivors who had their first intervention prior to the completion of 16 weeks of gestation. More encouraging outcomes following early treatment of patients with LUTO have also recently been published by other groups [66].

Oblivious to this progress, even the most recent consensus paper of the ERKNet CAKUT-Obstructive Uropathy Work Group does not provide any advice on how to proceed with very young LUTO fetuses between 12 to 16 weeks of gestation, despite quoting evidence from multiple references why and that, this group carries the poorest prognosis regarding renal function, pulmonary hypoplasia, termination rates, and postnatal survival [67].

During the review process and following the publication of our retrospective study, proponents of delayed VAS believed that most fetuses in our early group were overtreated and would have shown spontaneous resolution [68]. Furthermore, there was concern that the positive renal and pulmonary results we have seen in early life after early vesico-amniotic shunt insertion are unlikely to persist [68,69].

After more than three decades of therapeutic stagnation in this corner of prenatal medicine, I am personally convinced that the improved renal and pulmonary outcomes after early VAS in fetuses with LUTO will rekindle the interest in offering this procedure again—yet now far earlier in gestation. Nevertheless, I fully agree with any critic that further follow-up studies are required to elucidate the longevity of the more positive outcomes [68,69,70,71]. While male fetuses with a posterior urethral valve will benefit best from this strategy, improved survival of male fetuses with urethral atresia and female fetuses with uro-ano-genital malformations has also been observed. In both groups, complex postnatal reconstructive surgical attempts and urine diversion procedures can be expected.

### 2.10. Further Lifesaving Options for Fetuses with Renal Failure and Anhydramnios from Congenital Anomalies of the Kidneys and Urinary Tract (CACUT)

Far better chances of postnatal survival can be achieved nowadays in fetuses with terminal renal failure from urinary tract obstructions or renal agenesis. Only a decade ago, almost all fetuses diagnosed with renal anhydramnios before 20 weeks of gestation were doomed because they developed life-threatening lung hypoplasia over the course of gestation. When I learned in 2013 that peritoneal dialysis had become a standard therapy for treating neonatal kidney failure, I successfully began using FETO and the old approach of serial amnio-infusions as “lung rescue therapies” in order to prevent pulmonary hypoplasia and maintain its blood flow sufficiently high for survival [60]. Cameron and colleagues had already used serial amnioinfusions with the desired positive pulmonary effects in the early 1990s. They nevertheless concluded that “at present, this type of procedure is not an appropriate intervention in cases of renal agenesis, and such management is strongly discouraged” [72]. Their discouragement originated from the less successful postnatal management at the time when their patient who was born at 33 weeks of gestation died during postnatal treatment.

Fortunately, times have changed, and for the better; between the years 2016 until 2022, through serial amnio-infusions alone, we were able to rescue lung function in 17 of 20 fetuses (85%) with anhydramnios from renal conditions. After delivery, the infants were treated at cooperating institutions. In four, postnatal therapy was withheld because of non-pulmonary complications during their treatment course or additional malformations. To date, thirteen of them have survived to discharge and beyond (65%) and are either undergoing peritoneal dialysis or have already received a kidney transplant. None of them had been given any chance of survival by their primary diagnosticians.

These efforts also sparked the formation of two new important self-help groups in Germany, the *LUTO e.V.* and the *Bundesverband zur Begleitung vorgeburtlich erkrankter Kinder (BFVEK e.V)*. Both have become invaluable sources for affected families to receive and exchange information, hope, and comfort.

The risks and benefits of serial amnio-infusions in order to prevent the development of severe lung hypoplasia in fetuses with renal anomalies are also being examined in a large, ongoing multicenter *renal anhydramnios fetal therapy trial* (RAFT) (https://clinicaltrials.gov/ct2/show/NCT03101891) (Access date 20 December 2022) [73].

As a new area for research with multiple diagnostic and therapeutic ramifications for expectant mothers carrying fetuses at risk of pulmonary hypoplasia from anhydramnios, I would suggest some even less invasive interventions: Mindful counseling, relaxation, and anti-anxiety treatments, hypnosis, and perhaps also tocolytic agents. Based on the fact that fetal pulmonary hypoplasia in the presence of too little amniotic fluid mainly results from the upward displacement of fetal abdominal organs, the diaphragm, and the heart/lung base from fetal compression, any intervention that can alleviate maternal stress and anxiety, and, hence, fetal compression from increased maternal abdominal and uterine muscle tones may allow for better preservation of fetal lung size and blood flow over the course of gestation.

This mechanism provides a plausible explanation as to why an exceptionally serene mother carries a fetus with bilateral renal agenesis, that despite long standing anhydramnios presents with well-developed and perfused fetal lungs, a near normal four-chamber view of the heart, and a less cranially displaced heart–lung base at 25 weeks of gestation. These finding stand in stark contrast to the far more common observation where a usually chronically more anxious mother carries a fetus with considerably smaller lungs, pseudocardiomegaly, and a far cranially displaced heart–lung base at the same week.

### 2.11. Fetal Spina Bifida Surgery

Following the landmark randomized MOMS-trial that proved better neurological outcomes for prenatally operated patients [74], open fetal surgery for spina bifida is being carried out at many centers throughout the world. From the very beginning until today, the maternal invasiveness of the open surgical approach has been criticized from multiple angles [75,76].

Through trailing after the first unfortunate attempts of minimally invasive fetoscopic spina bifida closure by Bruner and Tulipan [77], it was my research group that succeeded step-by-step in developing and introducing a first fully percutaneous fetoscopic approach [78,79,80,81,82,83,84,85,86,87] (Figure 8). Following years of studies in sheep, inanimate models, and postmortem procedures—and to the present day close to 270 fetuses operated on since September 2002—it has become the preferred option in Germany for operating on unborn babies with spina bifida.

I was also able to teach my approach and support the clinical introduction of fully percutaneous minimally invasive fetal surgery for SBA in Brazil, Turkey, and Poland [88,89,90,91]. A growing number of centers—most of them united now under the umbrella of the “International Fetoscopic Neural Tube Defect Repair Consortium”—have introduced fetal surgery programs, which has increased the variety of minimally invasive surgical approaches for spina bifida [92]. As a result, I expect that open fetal surgery for this indication will soon become obsolete and be replaced by the minimally invasive approaches.

Emulating the original fetoscopic approach and following in the footsteps of Joseph Bruner and Noel Tulipan [77] and our sheep studies [78], some groups employ a hybrid approach that permits direct trocar insertion into a uterus that has been exteriorized by maternal laparotomy [93]. This setup permits plication of the chorioamniotic membranes at the site of trocar insertion. This maneuver has helped to substantially lower the risks of early amniotic fluid leakage, resulting in a mean prolongation of pregnancy of about two weeks compared to other fetoscopic approaches that forgo membrane closure. Fortunately, beyond 30 weeks of gestation, the clinical benefits of later delivery become less important, since no major differences in the morbidity from prematurity are to be expected. Since a membrane plication technique for the fully percutaneous approach is in reach, it remains my most preferred option. As one further alternative that combines the advantages of percutaneous fetoscopy and the hybrid approach, percutaneous/mini laparotomy fetoscopic repair may be employed [94].

### 2.12. How to Close Fetal Spina Bifida?

The proponents of open fetal surgery for spina bifida kept declaring time and again that the postnatal multi-layer closure approach ought to be the “gold-standard” or the “benchmark” of prenatal therapy, too. Yet countless repetition creates no truths. In fact, this way of thinking has shrouded the more obvious truth that the postnatal approach was never developed for fetal patients in the first place.

After open fetal surgery for spina bifida, patients showed only a little better leg function than a group of prenatally unoperated ones with similar lesion levels and types. In fact, the pioneers of fetal surgery for spina bifida published a paper summarizing their disappointment in their title: “Late gestation intrauterine myelomeningocele repair does not improve lower extremity function” [95].

After open fetal surgery for spina bifida, only one third of the MOMS patients showed a level of function that was two or more levels better than expected according to the anatomical level [74]. This outcome stands in stark contrast to our experience with fully percutaneous minimally invasive surgery, where at 30 months of life, the mean functional level was more than two segments better in about half of patients [96].

The analysis became even more impressive when focusing on high lesion levels between Th11 and L3 only. In this subgroup of 23 fetuses, more than half exhibited a functional level that was at least three segments better than the anatomical level. Of the whole study cohort, over 80% could walk, and independent ambulation without any aid was observed in about half of patients. Only one patient exhibited poorer leg function than what could be expected from the anatomical level [96]. Better preserved leg function after fetoscopy and poorer function after postnatal surgery were observed by independent investigators even when comparing my earliest patients with a cohort of prenatally unoperated patients [97].

I assume that the observed differences in leg functions are the consequence of the different handling of neurological tissues during the respective operative approaches. During multilayer closure performed in open fetal surgery, many sutures and knots are placed right adjacent to the neural cord [74]. These maneuvers already increase the risk of injury to the spinal tissue. Further damage from pressure-induced ischemia and deformation of neural tissues may result from the considerable downward compression of the spinal cord following direct skin closure. The disadvantage of this usually pursued surgical goal may play out even worse when the pedicles of the malformed vertebrae are short, resulting in a shallow spinal canal with the dorsal face of the neural cord lying at or even above skin level.

Unfortunately, some groups promoting fetoscopic surgery for spina bifida have bowed to the pressure to replicate similar multi-layer closure procedures [92,93]. Given the technical restraints and the small size of structures, these fetoscopic procedures are associated with an even higher risk of injury, not only to the spinal cord but to other anatomical structures [98].

In contrast, during our minimally invasive fetoscopic approach, the spinal tissue is carefully dissected from the surrounding structures and then simply covered water-tightly by a patch [83]. The patch is trimmed in such a fashion that it permits sufficient accumulation of cerebrospinal fluid between the neural tissue and the inner patch surface, decreasing the risk of adhesions and avoiding compression (Figure 9). In their assessment of the effect of open fetal surgery and minimally invasive fetoscopic surgery for spina bifida on neuromuscular outcomes, Verbeek and colleague observed in a small series of fetuses that segmental neuroprotection (sensory and motor function) was better preserved by fetoscopy than open fetal surgery (fetoscopy: +2 segments (−1.5–5) versus open: +0.25 segments (−2.5–6), respectively); (*p* = 0.04) [99].

That a similarly less invasive technique for fetal SBA coverage may allow a better preservation for the placode than the classic neurosurgical technique has also been demonstrated by seminal animal experiments comparing the two approaches by Silvia Herrera and colleagues [100]. I fully agree with the authors that “the current technique used for the correction of spina bifida (during open fetal surgery; and to my opinion even during postnatal surgery) in humans need to be reassessed”.

### 2.13. New Technique for Shunt Placement in Severely Hydropic Fetuses with Hydrothorax

Bilateral fetal hydrothorax may result in severe cardio-pulmonary compression with subsequent cardiac failure, ascites and massive skin edemas. Hydropic fetuses are at risk of dying from progressive decreases in fetoplacental blood flow, aggravated by placental dysfunction. In some cases, even the mother develops hydropic changes, mirroring the fetal disease (“Mirror”-Syndrome). In this situation, bilateral fetal thoraco-amniotic shunting is a lifesaving approach for about 50% of hydropic fetuses. Most commonly, double-pigtail catheters have been inserted into either chest cavity. Depending on fetal lie and activity, intrathoracic needle insertion and the placement of the external pigtail end may require extensive manipulations, thus increasing the risk of the premature rupture of membranes. Given the limited length of most devices, the procedure may be even more difficult or technically impossible in cases with massive skin edema (Figure 10).

In order to overcome the limitations of the current techniques, bilateral thoraco-amniotic drainage from a single insertion site is possible without the need for manipulation of fetal position (Figure 10). It puts little strain on the uterine wall and membranes and does not require amniotic fluid augmentation. The procedure is performed under materno–fetal anesthesia and usually lasts only a few minutes. All of the materials are widely available in hospitals with adult and pediatric intensive care.

In short, following a small maternal skin incision, an 18-gauge T-fastener needle is inserted dorso-laterally into the lower third of the fetal chest cavity that is nearest to the anterior uterine wall (Figure 10). After unloading the T-fastener and securing the fetal position by fixation of its suture, the needle is advanced closely above the diaphragm—crossing the midline between the thoracic aorta and esophagus—into the contralateral chest cavity. Then, a 0.035′′ guide wire is inserted through the needle shaft, followed by removal of the needle. The wire serves as a rail for the insertion of an 8-F-catheter sheath. Via the sheath, an adequately trimmed single pigtail catheter (about 5–7 cm) is placed across the midline such that it connects both chest cavities. Then, the sheath is pulled back into the proximal chest cavity. Here a second catheter is placed, extending from the proximal chest cavity into the surrounding amniotic cavity and uterine wall. Following passive drainage of the bilateral effusions via the catheter sheaths and catheters, the catheter sheath is removed without further manipulation and the incision is closed with a single stitch. As soon as the fetus awakens from anesthesia and begins to move again, it pulls the distal end of the catheter into the amniotic cavity, thus completing the procedure.

Following delivery, the external end of the catheter can be employed for further drainage or serve as a rail for exchange with a new catheter. The midline crossing pigtail and the T-fastener remain in the chest and can be removed electively by thoracoscopy around two to three years of age.

A last word of warning: As long as fetal hydrops is present, do not administer steroids to the mother for enhancing fetal lung maturation. In this situation, steroids may facilitate the development of maternal peripheral edema, pleural effusions, and pulmonary edema (iatrogenic “mirror syndrome”).

### 2.14. Materno–Fetal Hyperoxygenation for Fetal Heart Malformations

Inspired by the excellent book *Fetus and Neonate – Physiology and clinical applications* by Hanson, Spencer, and Rodeck [101] during my formative research fellowship at UCSF from 1993 to 1996, I had been tinkering with the use of oxygen in my mind and had suggested it to the deputy chief of fetal surgery as a non-invasive tool for the assessment of disease severity in fetuses with diaphragmatic hernia. Unfortunately, I let myself be discouraged at the time.

A few years later, I briefly met Juha Rasanen during his research fellowship at Jim Huhta’s fetal echo laboratory in Philadelphia. Rasanen’s clinical studies provided the first proof that the pulmonary vasculature of human fetuses indeed responds to materno–fetal hyperoxygenation with vasodilation [102]. I also learned that the Philadelphian team was successfully using diagnostic hyperoxygenation for the assessment of disease severity in fetuses with diaphragmatic hernia.

Yet it was not until 2002 that I could gather my own practical experience with diagnostic materno–fetal hyperoxygenation in fetuses with this condition and noticed the acutely enlarging loading effect on the commonly smaller left heart structures. It took me another five years, in 2007, after about 15 years of immersing myself deeply into the study of the fetal heart diseases and the development of fetal cardiac interventions, when I envisioned the potential of chronic intermittent materno–fetal hyperoxygenation as a non-invasive means for achieving the catch-up growth of hypoplastic cardiac structures; in my first case, aortic coarctation [103].

The principle is simple: A small fraction of supplemental oxygen being inhaled by pregnant women beyond 30 weeks of gestation crosses the placenta and increases the oxygen content of the fetal blood, thus inducing fetal pulmonary vasodilatation and a marked increase in lung blood flow (Figure 11). Through this mechanism, the preload from the increased pulmonary venous return challenges both hypoplastic left and right heart structures (Figure 11). Through providing 6 L/min of O_2_ via a face mask to the mother (approximately 35–40% oxygen) three times daily (8:00–12:00/14:00–18:00/20:00–23:00; total 10–11 h/day) from 32 + 0 to 34 + 0 weeks of gestation onwards until delivery, impressive degrees of catch-up growth can be observed in suitable cases [103].

Despite its enormous potential for fetuses with a wide spectrum of cardiovascular malformations, doubts were raised about the efficacy of the approach. It was feared that the increased pulmonary venous return would interfere with the normal right to left shunt across the oval foramen toward the left side of the heart, neutralizing additional loading of the left side of the heart. Whereas such a mechanism may be present in some fetuses, it is not at all applicable to fetuses with an imperforate atrial septum and an otherwise structurally normal heart. This subgroup presents with severe left heart underdevelopment (z-scores ≤ −3) and in my experience benefits particularly well from the volume challenge provided to the left side of the heart by chronic intermittent materno–fetal hyperoxygenation [104].

In other fetuses with fetal left heart hypoplasia, abnormal, preferential streaming of ductus venosus blood toward the right side of the heart can be observed [105]. This flow abnormality not only results in decreased flow across a patent oval foramen but is also associated with lower pulmonary blood flow. Both factors contribute to the development of left heart hypoplasia and may be ameliorated by materno-fetal hyperoxygenation.

Following my initial publication of a small series of non-controlled fetuses at risk of coarctation or hypoplastic left heart complex [103], the approach has received world-wide attention over the past decade [106,107,108,109,110,111]. The important question—apart from the efficacy of the new therapy—is: will there be untoward consequences—in particular on the fetal brain and, as a consequence, on postnatal neurological development—in the short and long run? In a small study with multiple limitations, Edwards and colleagues observed small decreases of biparietal diameter z-scores in fetuses over the course of materno–fetal hyperoxygenation [108]. After delivery, no difference in neurodevelopmental testing could be observed between prenatally treated infants and prenatally untreated controls. A puzzling finding of the same study was that fetuses that underwent hyperoxygenation for longer than nine hours per day and with less interruptions—as suggested in my original protocol—even exhibited improved head growth.

From the opposite perspective, other authors suspected reduced oxygen consumption in fetuses with left heart malformations to be the cause of smaller brains compared to healthy fetuses [109]. They concluded that impaired brain growth in fetuses with these lesions “raises the possibility that their prenatal brain development could be improved by maternal oxygen therapy”. Their conclusion received strong support by a recently published prospective longitudinal study; fetuses with left-sided obstructive cardiac lesions had significantly larger head circumferences and increased intracranial volumes after chronic hyperoxygenation compared to those that did not undergo this therapy [110]. In a meta-analysis, Jeniver CoVu and coauthors also did not find evidence of serious adverse effects from materno–fetal hyperoxygenation [111]. These observations encourage further exploration of this promising new approach.

Already in my first paper, I stated that “given its simplicity, universal availability, and potential benefits for large numbers of patients, intensive research at dedicated centers is now required with the goals to optimize hyperoxygenation schedules, define all suitable cardiac malformations, and assess the short- and long-term safety of this novel therapeutic approach”. Thus, I was happy to learn that Zeng and colleagues performed the first prospective randomized trial employing maternal hyperoxygenation in fetuses suspected at risk for coarctation. In their study, the rate of postnatal surgery for coarctation was dramatically lower in the treated group (20%) than in the untreated control group (75%) [112]. Furthermore, ventricular function improved over time in the treatment group [113]. Again, no untoward effects of materno–fetal hyperoxygenation were observed [112].

Recently, I observed that chronic intermittent materno–fetal hyperoxygenation can also effectively be used in fetuses with hypoplastic left heart syndrome and aortic atresia. Here, the approach promotes higher amounts of retrograde flow across the fetal aortic arch into the ascending aorta and from there into the coronary arteries to meet the needs of the increased cardiac workload. As a result of prenatal hyperoxygenation, extremely hypoplastic ascending aortas—a known risk factor for postnatal palliative surgery in neonates with hypoplastic left heart syndrome—as well as the aortic arch become larger, which may facilitate postnatal treatment (Figure 12). Another advantage of chronic hyperoxygenation in fetuses with hypoplastic left heart syndrome is that this subgroup benefits particularly well by achieving significantly larger head circumferences and increased intracranial volumes compared to those that did not undergo this therapy [110].

With generous financial support from former patients, private donors, and the Hector foundation (Weinheim, Germany—www.hector-stiftung.de) (Access date 20 December 2022), our group is now in the final stages of preparing the prospective, randomized, multicenter trial “HYPEROX”. Our goal is to assess whether any hypoplastic fetal left heart structures may benefit from the approach and whether it is possible to predict which structure(s) is/are thought to benefit the most from it even before starting therapy. Furthermore, more scientifically sound and robust data about its maternal and fetal risks and safety will be generated.

### 2.15. Fetal Cardiac Interventions and Resuscitation

Severe fetal semilunar valve obstructions observed in mid-gestation commonly destroy their associated left or right ventricles until full term. As a result, only palliative surgical or interventional approaches can be offered to the affected patients. For life, they have to get by with the progressively accumulating side effects and risks of a univentricular circulation. In order to interfere with this detrimental natural course of the obstruction, Maxwell, Allan, and Tynan were the first to offer fetal balloon valvuloplasty for this condition [115]. In highly selected cases, the procedure may aid in preserving the function of the affected left or right side of the fetal heart [116].

Unfortunately, fetal balloon valvuloplasty is—even in experienced hands—still associated with a 10% risk of fetal demise [117,118]. This deeply frustrating event is usually the consequence of sustained bradycardias or complete cardiac arrest, commonly associated with hemopericardium. Furthermore, direct injury from the intervention needle into the cardiac conduction system has been implicated in some fatal cases, as it is situated adjacent to the left ventricular outflow tract and runs immediately proximal the aortic valve.

Once cardiac access has been achieved, an epinephrine or atropine bolus into the fetal heart can be given as prophylaxis against sustained fetal bradycardias and cardiac arrest during fetal cardiac intervention. Yet in a few cases where life-threatening prolonged bradycardia or arrest has occurred, I found direct fetal cardiac massage to be the only effective technique for successful resuscitation [119].

In order to achieve this feat, the transducer rhythmically compresses the maternal abdomen over the fetal chest and heart at a rate of about 100 beats per minute. The correct position of the transducer, the necessary depth of the compressions, and the effect of the cardiac massage on the fetal heart and fetoplacental circulation can immediately be observed in real time: (https://www.youtube.com/watch?v=7IBn7cWZrJo) (Access date 20 December 2022). As in the postnatal situation, it is important, to provide sufficient time for resuscitation and not stop too early. In the fetus, shown in the video, the technique was carried out over 20 min before contractions sufficient for fetal survival re-occurred.

A similar approach by digital resuscitation was first published in 1984 by Kypros Nicolaides and Charles Rodeck in a hydropic 19-week-old anemic fetus with Rh-isoimmunization when cardiac arrest occurred during fetoscopic blood transfusion [120]. As in our case, no detrimental fetal brain damage or other injuries have been observed following the application of direct fetal cardiac massage. Furthermore, the approach was safe for the mother and did not result in the rupture of membranes or placental abruption.

After reading these sections and watching the video, any interventionist confronted with prolonged, life-threatening fetal bradycardias during fetal cardiac or non-cardiac interventions may now readily consider this option.

### 2.16. Not an Easy Path…

The performance of the best possible interventions and in particular the introduction of new approaches requires great courage, frustration tolerance, and endurance. Serious fetal interventionists need to stomach the fact that despite their best efforts and intentions, about 10–40% of their frail patients will die. Death results from a multitude of causes may come early, right at or shortly after fetal intervention, throughout the course of gestation, upon delivery, or during the postnatal treatment course.

The impact of this dire situation may be aggravated when additional toxic criticism is mounted on fetal surgeons when interventions fail technically or fetuses do not benefit from attempts at their salvage.

This criticism is expressed not only by individuals with no or little experience of performing fetal interventions, but more often from competing insiders. Particularly from this latter group, published evidence—of course by at least one randomized trial—is aggressively being demanded, even for first-in-man-procedures or for lifesaving treatment attempts in extremely rare and complex cases; demands that are impossible to be met by anyone.

Yet, even seasoned investigators can be completely oblivious or ignorant to the fact that apart from experienced interventionists, adequate selection criteria, treatment plans, save and reproducible surgical techniques as well as functional devices are necessary prerequisites in order to assess the full clinical potential and risks of any intervention in a randomized trial. The planning fallacy of examining a faulty intervention approach was only one of several central problems of the PLUTO trial in fetuses with severe lower urinary tract obstructions [59]. Its widely disseminated results have haunted the further development of lifesaving vesico-amniotic shunt insertion ever since its publication. The same mistake may now be repeated by pushing too early for randomized trials in other conditions.

Given the required formalities of clinical studies and medical device development studies according to the latest directives of Good Clinical Practice (GCP) and the Medical Device Regulation (MDR), the costs for clinical studies, device development and certification easily reach several of millions of Euros. As funding for investigator-initiated trials in the arena of fetal surgery is extremely hard to obtain, a united push by leading fetal interventionists is urgently required to lobby for nuanced deregulation and adequate financial support.

Fortunately, most obstacles can be overcome with stamina, patience and some lucky streaks. In addition, their stings pale in comparison next to the deep gratitude and joy with which fetal interventionist and their postnatal colleagues are rewarded by the families whose lives they have changed forever and in a good way.

### 2.17. Another Hidden Mortality

Well into the 1980s, a pediatric surgeon’s perspective on the mortality of a neonate with a diaphragmatic hernia was optimistic, considering that the majority of the ones they actually were able to operate on would survive. In contrast, Michael Harrison put a huge damper on this optimism, when he observed that most neonates born with this malformation never made it into the operating room but died early after their delivery. This finding affirmed him in his quest to develop lifesaving fetal surgical procedures [4].

In addition to the hidden mortality, as described by Harrison, I have observed a second kind of hidden mortality that has not been previously considered, and is much harder to fathom or count: the many fetuses that would have been eligible for fetal interventions, but had died because they were not diagnosed correctly or were diagnosed too late, or were rejected because they were assumed to be not yet ill enough or assumed of being too ill, or those who were not referred by doubting, uninformed, or simply ignorant practitioners. All of these cases remain unaccounted for. It can be assumed that over the past 20 years in Germany alone, several hundred potentially avoidable fetal deaths were the result of these circumstances. It is likely, that here there have been more deaths on their account than in the context of developing and performing fetal surgery.

In order to improve upon this complex situation in my country, I have undertaken various steps: firstly, with the help of a cohort of my patients in 2017, I initiated the first German self-help group for families suffering the nightmare of the prenatal diagnosis of a grave condition in their unborn babies. The self-help group BFVEK (www.bfvek.de) (Access date 20 December 2022) provides instant help, guidance, and other important resources to support affected women and their families. In addition, accompanied by a group of veteran lawyers, ethicists and prenatal medicine specialists, we are working on a framework for patient counseling in the context of fetal diagnostics and therapy and to aid in informing medical advisors about their obligations. I hope that these steps will aid affected expectant women to obtain better access to established procedures and the latest developments in the field of fetal interventions.

### 2.18. Quo Vadis, Fetal Surgery

Fetal surgery, postnatally followed and complimented by the great therapeutic powers of modern neonatology, has become a lifesaving reality for hundreds of fetuses each year. The time has come to solicit for the appropriate academic recognition of the matured field as an independent specialty in order to provide the best possible care of fetal patients. Existing medico-legal obligations to refer potential fetal patients to specialist centers only and the prevention of expectant mothers and fathers from being counseled wrongly need to be reinforced. These steps will allow more affected expectant women and their unborn children to obtain access to the lifesaving or quality-of-life-improving possibilities of modern minimally invasive fetal surgery and therapy. The opportunity to treat more patients at dedicated centers will also result in more opportunities for the research of rare diseases and conditions, promising even better pre- and postnatal care in the future.

## Figures and Tables

**Figure 1 children-10-00067-f001:**
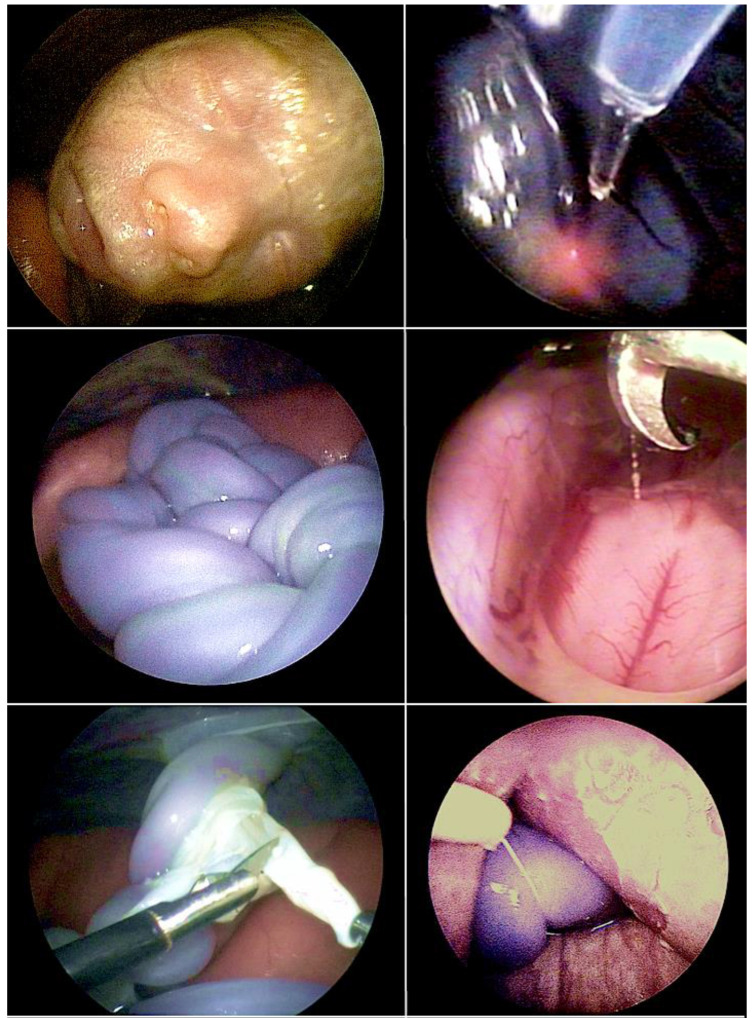
The technique of fully percutaneous partial amniotic carbon dioxide insufflation (PACI) has been invaluable in improving the visualization of both the fetus (**top left**) and intraamniotic contents during complex fetoscopic procedures. Over the past 20 years, I have employed it for fetoscopic laser ablation for twin-to-twin transfusion syndrome (TTTS) (**top right**), umbilical cord transection in monoamniotic twins (**middle left**), during surgery for spina bifida (**middle right**), amniotic band removal (**bottom left**), and during umbilical cord ligation in discordant twins (**bottom right**).

**Figure 2 children-10-00067-f002:**
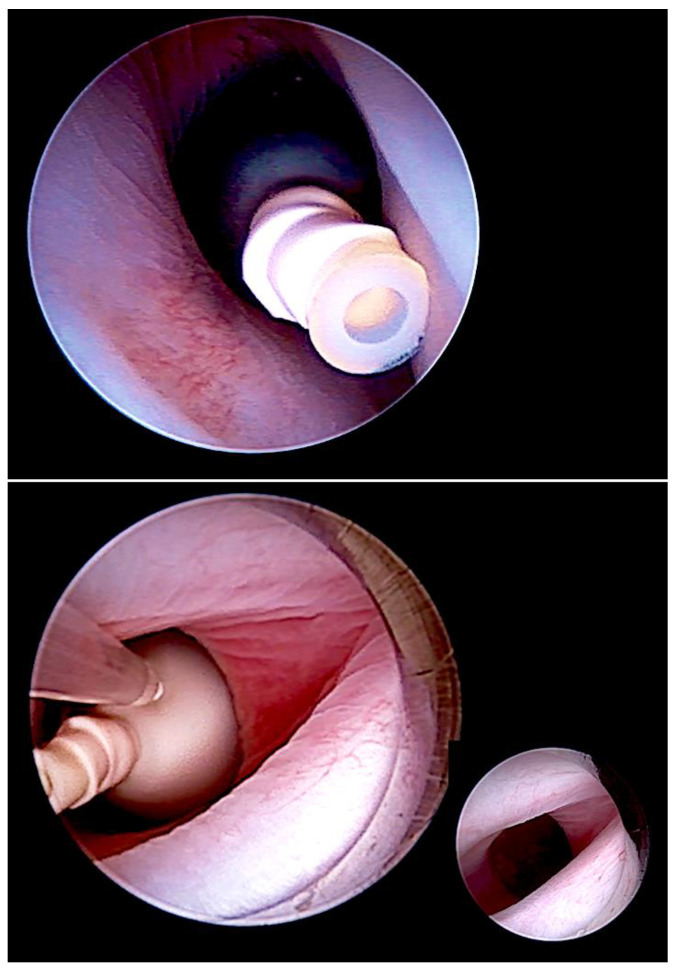
Minimally invasive, fully percutaneous fetoscopic tracheal balloon occlusion (FETO) (**top**) for the treatment of fetuses with severe pulmonary hypoplasia. FETO has most commonly been used in fetuses with diaphragmatic hernia. In addition, we observed it to be also particularly effective for the treatment of lungs compromised by long-standing anhydramnios following early preterm premature rupture of membranes prior to 20 weeks of gestation. It can also be applied to save fetuses with pulmonary hypoplasia from poorly distending and poorly perfused lungs after drainage of severe hydrothorax or after volume reduction of CPAMS. Following a period of one to several weeks, the balloon is removed by a second intervention (**bottom**).

**Figure 3 children-10-00067-f003:**
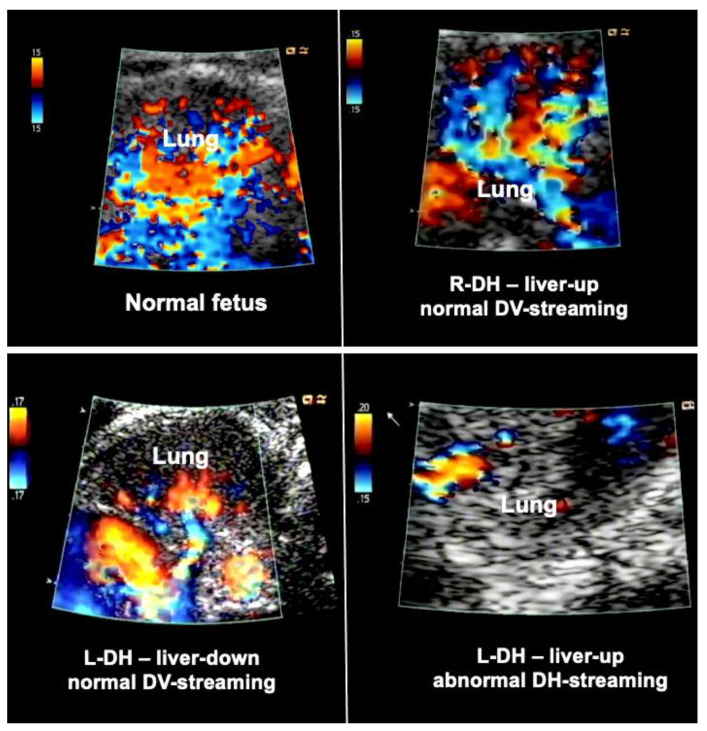
Examples of lung blood flows of a normal fetus (**top left**) and fetuses with diaphragmatic hernia during diagnostic materno–fetal hyperoxygenation. Most fetuses with liver-down left diaphragmatic hernia (L-DH) exhibit normal streaming of ductus venosus blood into the left side of the heart (**bottom left**). As a result, both sides of the heart are of equal size, pulmonary blood flow signals are easily visualized, and pulmonary vasodilation in response to diagnostic maternal hyperoxygenation can be produced. In contrast, most fetuses with liver-up L-DH exhibit preferential streaming of ductus venosus blood into the right side of the heart of the fetus. As a result, there is less preload to the left side of the heart, which in turn becomes hypoplastic. In addition, pulmonary blood flow is often markedly decreased (**bottom right**). Fetuses with right diaphragmatic hernia (R-DH), despite liver herniation, present with normal ductus venosus streaming into the heart. Therefore, as a rule, in these fetuses, neither isolated left, nor isolated right heart hypoplasia are observed, and color Doppler signals of pulmonary blood flow are often normal (**top right**).

**Figure 4 children-10-00067-f004:**
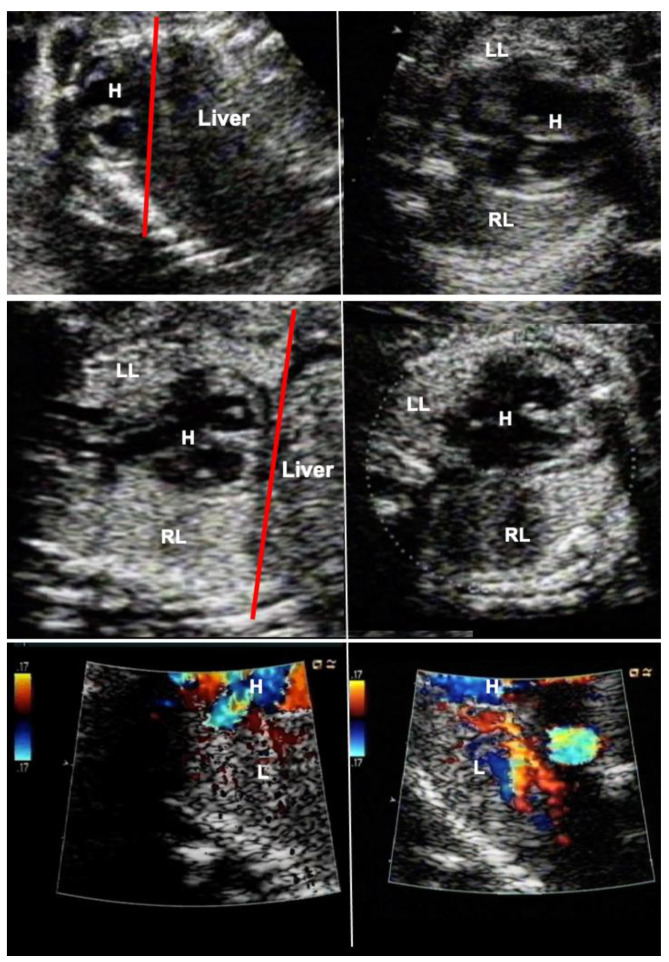
Effect of FETO at 27 weeks of gestation in a fetus with severe pulmonary hypoplasia from anhydramnios after rupture of membranes at 15 weeks of gestation. **Top left**—severe compression of the fetus has resulted in a displacement of the heart–lung base to the level of the third to fourth rib (red line). **Top right**—there is pseudocardiomegaly and at the level of the four-chamber-view, the left lung (LL) can hardly be seen. **Middle left**—after one week of FETO, the lung has regained volume such that the heart–lung base now lies at the level of the 7th–8th rib (red line). **Middle right**—as a result, the four-chamber-view has normalized, and the left lung can easily be seen. (RR—right lung). **Bottom left**—color Doppler imaging at a low Nyquist limit (17 cm/sec) demonstrates barely any color flow signals within the lungs one day after FETO. **Bottom right**—on the third day of FETO, color flow signals are clearly present in the fetal lung, heralding a better prognosis. As soon as fetal pulmonary vasodilatation can be provoked by diagnostic materno–fetal hyperoxygenation, the fetus is scheduled for fetoscopic balloon removal, delivery, and postnatal treatment.

**Figure 5 children-10-00067-f005:**
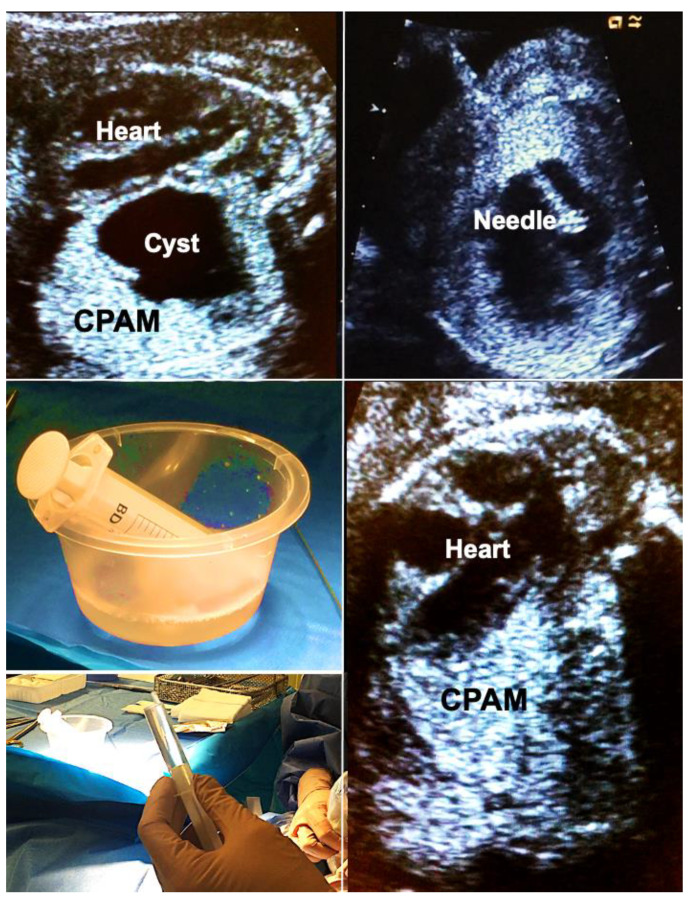
Alcohol ablation in hydropic fetuses with congenital pulmonary airway malformations (CPAM) permits immediate and sustained volume reduction of large, life-threatening malformations. **Top left**—large left-sided CPAM with contralateral displacement and compression of the fetal heart. **Top right**—an 18-gauge needle has been placed into a large solitary cyst of the CPAM. **Middle left**—40 mL of fluid were aspirated from the cyst. **Bottom left**—then, the cyst is filled with 5 mL of pure ethanol. **Bottom right**—after 5 min, the ethanol is aspirated. Immediate improvement in cardiac filling can be observed. Hydrops usually resolves within days. In this case, the remainder of gestation remained uneventful. Survival to discharge was achieved after postnatal surgical removal of the CPAM.

**Figure 6 children-10-00067-f006:**
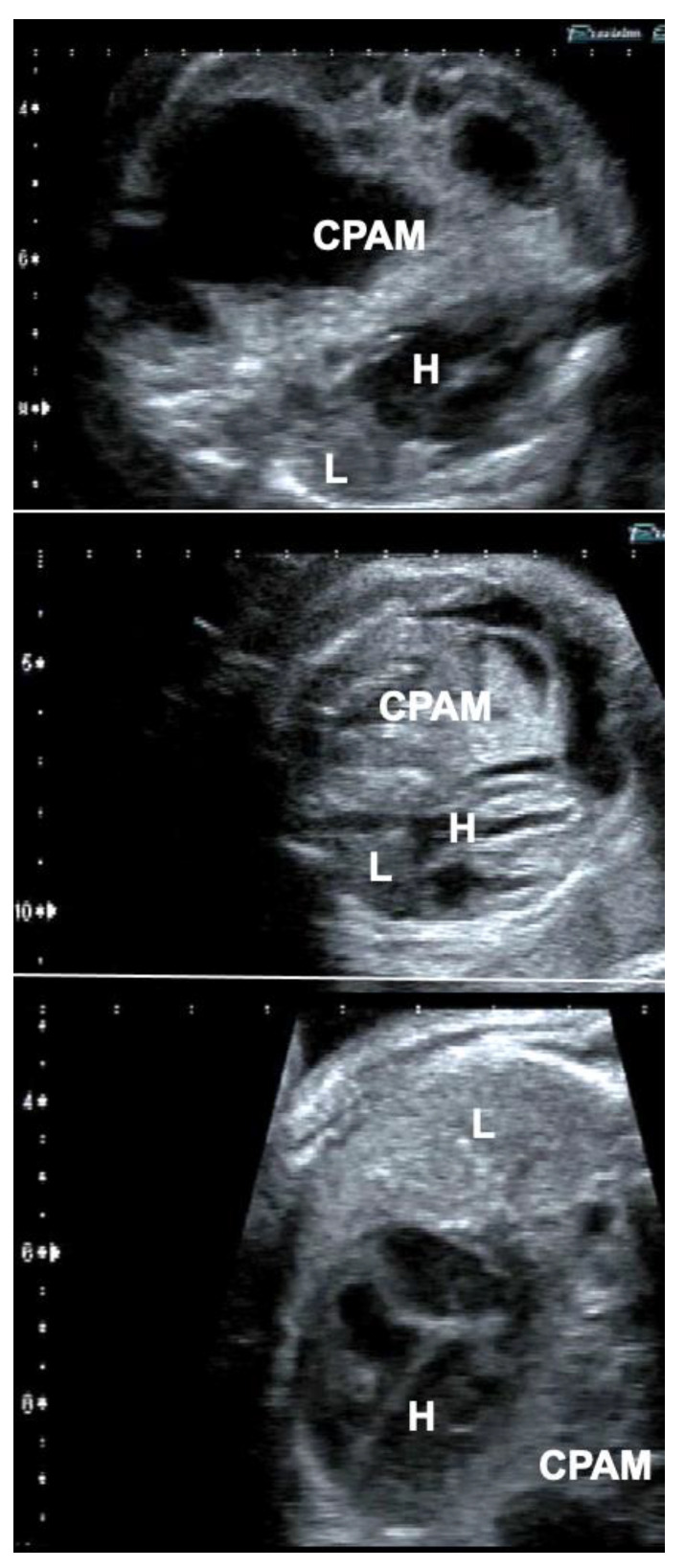
Effect of FETO in a fetus with a giant congenital pulmonary airway malformation (CPAM) (**top**), whose lungs did not distend—despite several previous interventions—after long-standing compression and were considered too small or too dysfunctional for postnatal survival (**middle**). In this case, FETO at 30 + 4 weeks of gestation served as the last resort to achieve lifesaving lung catch-up growth (**bottom**). H = heart; L = lung.

**Figure 7 children-10-00067-f007:**
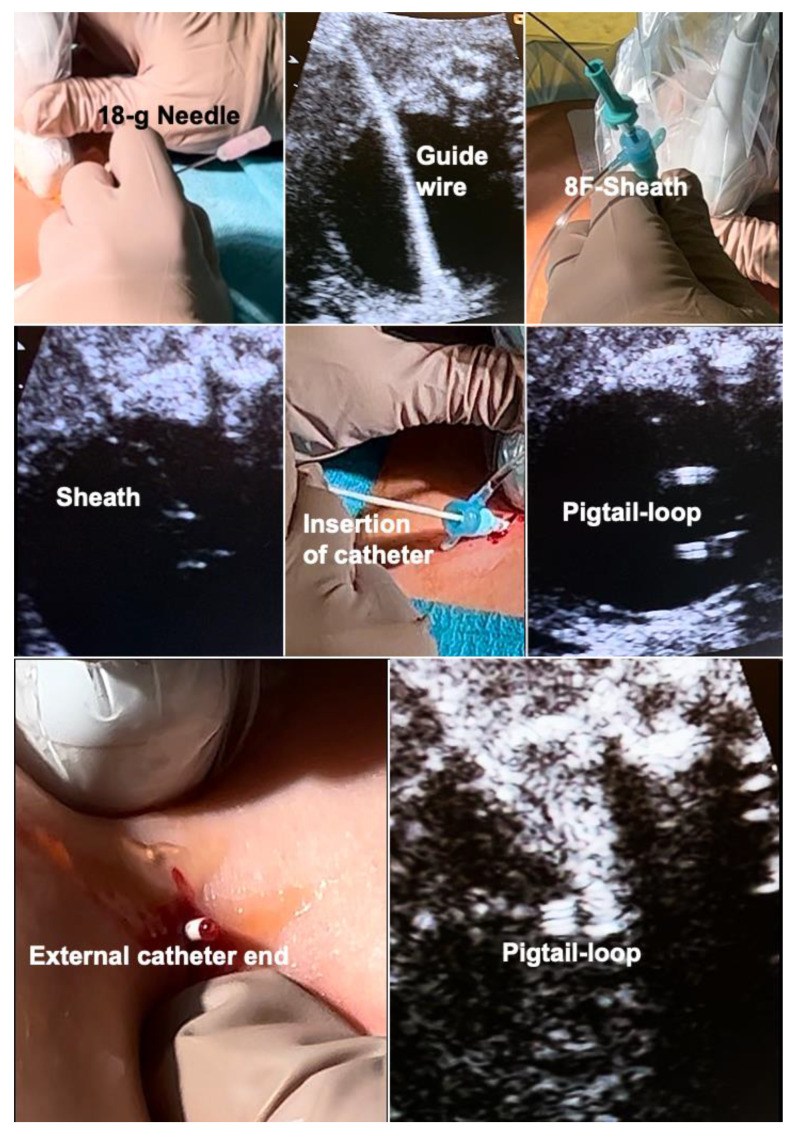
Novel shunt technique for fetuses with megacystis from lower urinary tract obstruction (LUTO) in order to achieve safe, early second trimester shunting. **Top left**—after a small skin incision, percutaneous ultrasound-guided puncture of the fetal bladder with an 18-gauge needle is performed. **Top center**—this is followed by insertion of a 0.035′′-guide wire via the needle shaft into the fetal bladder and—**top right**—insertion of an 8F-sheath—over the wire—into the fetal bladder. **Middle left**—the tip of the sheath can be seen inside the bladder. **Middle center**—A single-pigtail catheter has been mounted on the trocar of an 18-gauge needle and is being inserted via the sheath into the fetal bladder. **Middle right**—after removal of the needle trocar, the pigtail loop becomes visible. **Bottom left**—after removal of the 8F-sheath, fetal urine drips from the distal end of the catheter. Then, the catheter end is pushed below the skin level of the maternal abdomen and the skin incision is closed with a suture. **Bottom right**—ultrasound imaging after the procedure demonstrates the pigtail loop within the empty bladder.

**Figure 8 children-10-00067-f008:**
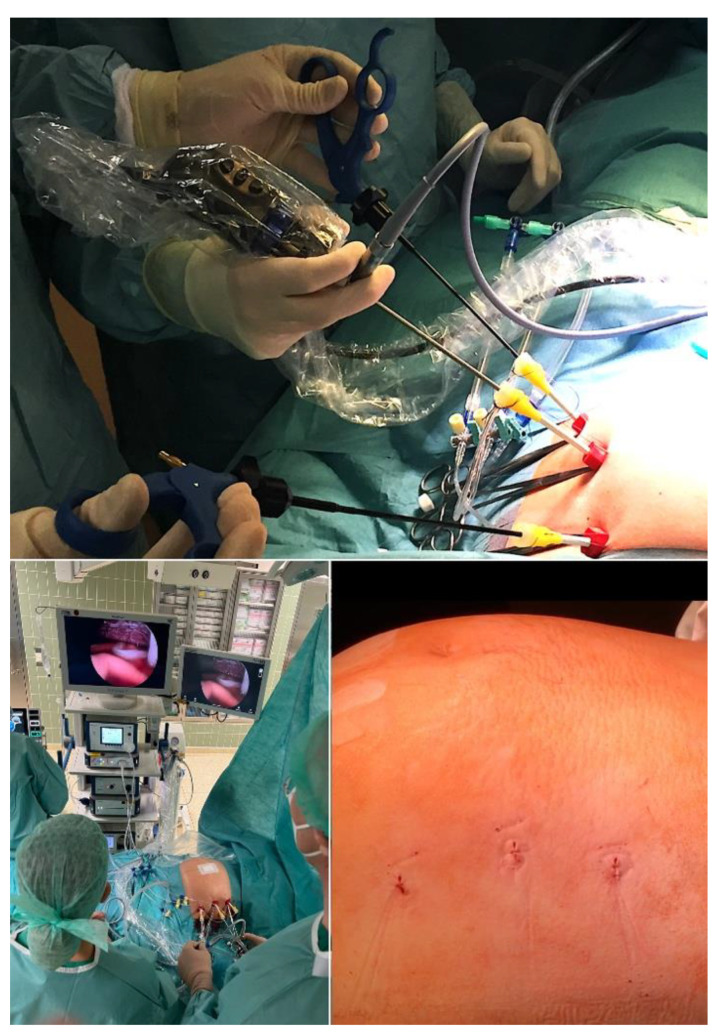
External aspect of the fully percutaneous setup for minimally invasive fetoscopic surgery for fetal spina bifida aperta. **Top**—after percutaneous intraamniotic insertion of three 11F-sheaths, followed by partial amniotic carbon dioxide insufflation (PACI), and fetal posturing (**bottom left**), surgery on the malformation (Figure 9) can commence. **Bottom right**—the maternal abdomen after trocar removal illustrates the minimal maternal trauma of the approach.

**Figure 9 children-10-00067-f009:**
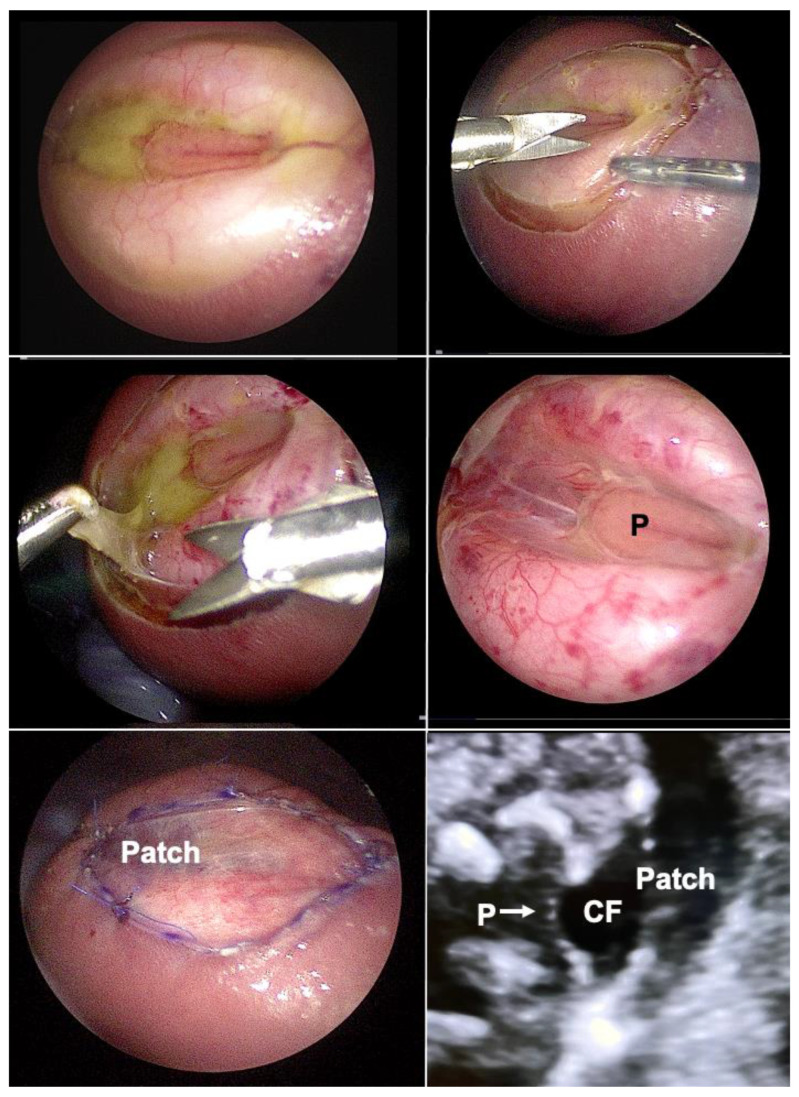
**Top left**—fetoscopic aspect of a flat myelomeningocele at the L4 level at 25 weeks of gestation after successful fetal posturing in the carbon dioxide-insufflated amniotic cavity. **Top right**—at first, the lesion is circumcised, then, the placode (P) and spinal nerves are carefully dissected from surrounding structures (**middle left** and **right**). The neural tissue is then covered water-tightly by a patch (**bottom left**). The patch has been trimmed in such a fashion that it permits sufficient accumulation of cerebrospinal fluid (CF) between the placode (P arrow) and the inner patch surface, greatly decreasing the risk for adhesions and avoiding compression (**bottom right**).

**Figure 10 children-10-00067-f010:**
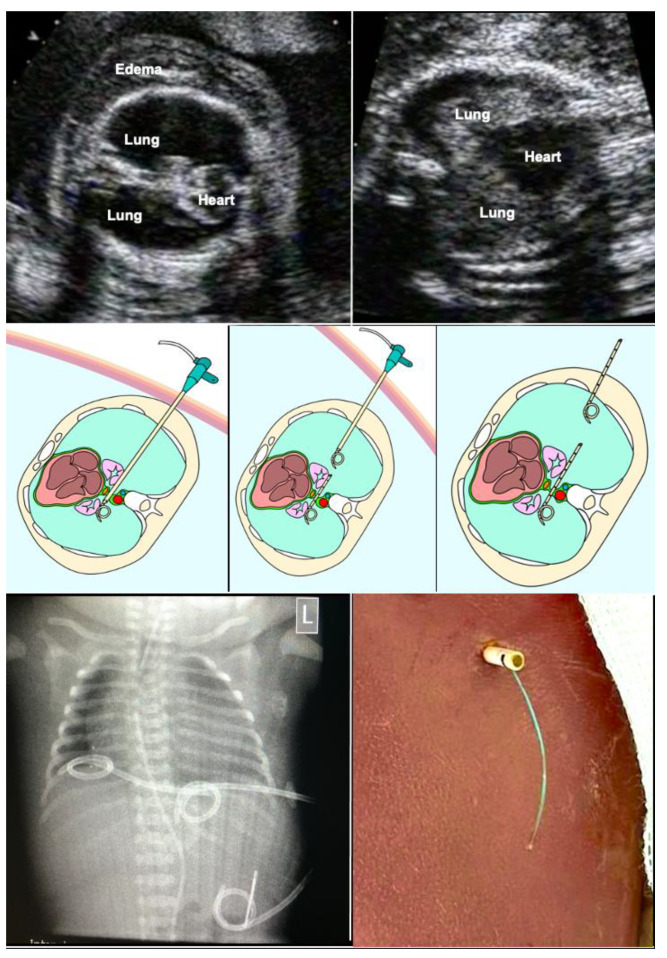
Bilateral thoraco-amniotic drainage from a single insertion site. **Top left**—before the intervention, bilateral hydrothorax with severe pulmonary and cardiac compression as well as marked skin edema are observed at 26 weeks of gestation. **Top right**—after the intervention, the lungs are distended again and cardiac filling has improved. Technique: **middle left**—without manipulation of fetal position or amniotic fluid augmentation, a catheter sheath is placed closely above the diaphragm from the proximal into the distal thoracic cavity. Via the sheath, an adequately trimmed single pigtail catheter (about 5 cm) is placed across the midline such that it connects both chest cavities. **Middle center**—then, the sheath is pulled back into the proximal chest cavity. **Middle right**—here a second catheter is placed, extending from the proximal chest cavity for about 6 cm into the surrounding amniotic cavity and uterine wall. Following passive drainage of the bilateral effusions via the catheter sheaths and catheters, the catheter sheath is removed without further manipulation and the maternal abdominal skin incision is closed with a single stitch. As soon as the fetus awakens from anesthesia and begins to move again, it pulls the distal end of the catheter into the amniotic cavity, thus completing the procedure.

**Figure 11 children-10-00067-f011:**
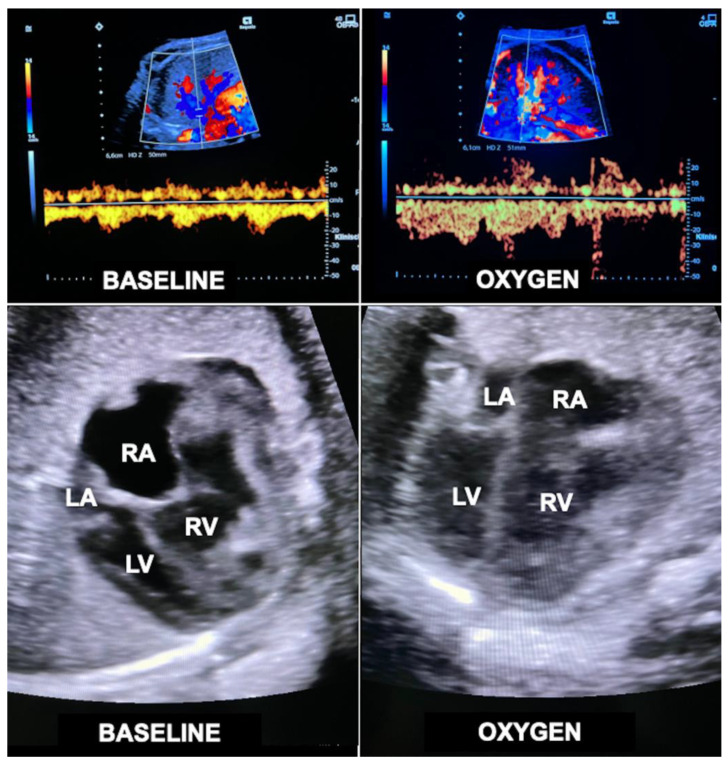
Acute loading effect of materno–fetal hyperoxygenation on the left ventricle in a fetus with hypoplastic left heart complex at 34 + 2 weeks of gestation. **Top**—color Doppler interrogation of a pulmonary vein before (**left**) and after 10 min (**right**) of materno–fetal hyperoxygenation demonstrates a marked increase in pulmonary flow. **Bottom left**—there is marked disproportion between the hypoplastic left (LV) and the dilated right ventricle (RV) before hyperoxygenation (BASELINE). **Bottom right**—after only 10 min of materno–fetal hyperoxygenation (OXYGEN), the left ventricle shows substantially improved filling at end diastole. Over several weeks, this effect results in catch-up growth of hypoplastic left heart structures. LA = left atrium; RA = right atrium.

**Figure 12 children-10-00067-f012:**
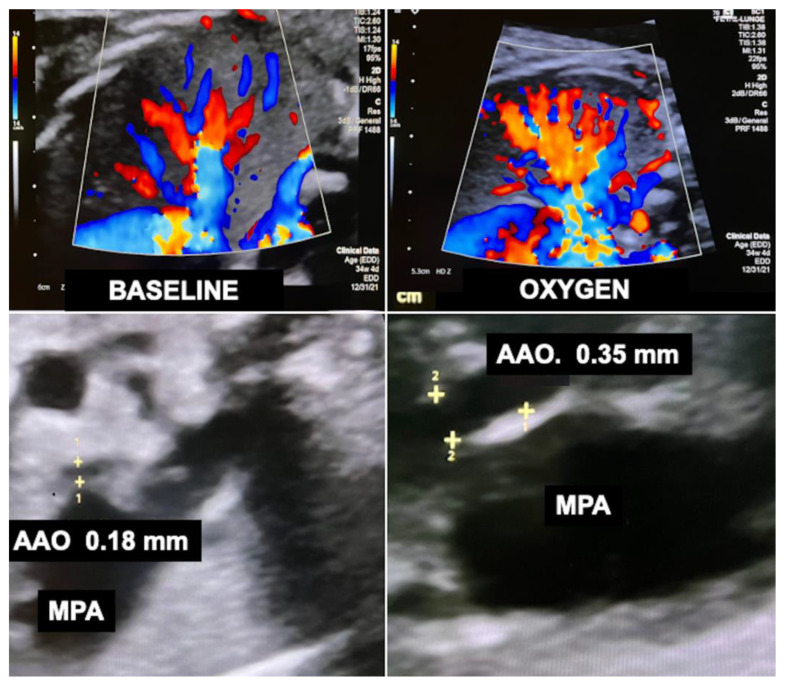
Effect of chronic intermittent materno–fetal hyperoxygenation (Kohl Procedure) on the development of a severely hypoplastic ascending aorta which serves in essence as “the common coronary artery” in a fetus with hypoplastic left heart syndrome. **Top**—color Doppler interrogation of the fetal lung before (**left**) and during (**right**) materno–fetal hyperoxygenation demonstrates a marked increase in flow signals. **Bottom left**—at the beginning of therapy at 34 + 4 weeks of gestation, adjacent to the much larger main pulmonary artery (MPA), the ascending aorta (AAO) is difficult to find as it measures only 1.8 mm (z-score −9.1 after Krishnan et al. 2016 [114]). Its small size constitutes a risk factor for the postnatal occurrence of cardiac ischemic events. During materno–fetal hyperoxygenation, the retrograde flow across the aortic arch and into the ascending aorta increases. **Bottom right**—after 30 days of daily therapy, the diameter of the ascending aorta has increased to 3.5–4 mm (z-score ≥ −6.2), reducing the risk of cardia ischemia and facilitating the performance of postnatal interventions.

## Data Availability

No new data were created for writing this manuscript.

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
