# Peer review of "Lifesaving Treatments for the Tiniest Patients—A Narrative Description of Old and New Minimally Invasive Approaches in the Arena of Fetal Surgery"

_children, 2022, doi:10.3390/children10010067_

Round 1

Reviewer 1 Report

Dear Author,

The presented study is a narrative description about old and new minimally-invasive approaches in the arena of fetal surgery. This is essentially a summary of a new minimal-invasive approaches in fetal surgery and as such will be welcomed by both medical students and practitioners. I have read the article with a great interest. The author has been successful in his effort to correlate pathology and physiology with the clinical and therapeutic aspects of diseases and throughout the article emphasized the necessity of further investigations. In addition to a section on " Fetal cardiac interventions and resuscitation," there has been included a wealth of modern material on cardiac anomalies treatment.

Author Response

Thank you for your review and for your kind comments regarding my manuscript!

No changes requested.  

Reviewer 2 Report

I have been asked to peer review “Life-saving treatments for the tiniest patients- A narrative description about old and new minimally-invasive approaches in the arena of fetal surgery” by Professor Thomas Kohl, an expert in the field of fetal surgery. Great topic, and a great article for maternal-fetal specialists! The manuscript is very well written, very easy to follow ,  focused on fetoscopic tracheal balloon occlusion for diaphragmatic hernia and in fetuses at risk for severe pulmonary hypoplasia from anhydramnios, specific “in utero” treatments for congenital pulmonary airway malformations,  lower urinary tract obstruction and renal agenesis,  spina bifida, hydrothorax and hydrops fetalis , and fetal semilunar valve obstructions. The Author shares his opinions,  and his techniques  in fetal surgery over the course of his career who is contemporary with the description of the natural history of these diseases, the development of new pathologies, new techniques and new devices. Also, the multidisciplinary approach and different protocols from different countries because fetal surgery, including accurate prenatal diagnosis and postnatal recovery is expensive.   

Some observations :

Lines 98-100, Lines 178-180: In my opinion is beyond the subject .

Line 207 : about PACI – I suggest to describe the technique

Line 378 : CPAM congenital pulmonary airway malformation

Reference 6  is the same as reference 7 .

Author Response

Thank you for your review and for your kind comments regarding my manuscript.

Despite your opinion, I have left the lines 98 to 100 and 178 to 180 of the original manuscript. Nevertheless, I have revised the lines for more clarity.

Of course, falsely claiming to be the inventor of novel procedures is nothing new in medicine and any other field. Yet, the truth has to prevail. And that is my intention.

The problem of a lack of sufficient hygiene in fetal intervention is huge. The risks of chorioamnionitis and fetal infection to this day remain primary issues for voting against fetal surgery. It will be mind blowing for many readers to see that fetal interventions are taught and carried out in a kind of living room scenario. And that the practitioner seen here is the leader of the biggest international fetal therapy institution is nothing less but starkly bewildering. As a leading developer of safe and reproducible techniques for fetal therapy, I cannot help but highlight this problem in my manuscript to the widest possible audience.

Line 207 – According to your suggestion, I have provided a very detailed description of how to perform amniotic insufflation, and provided information about all important points that need to be taken into account for its safe application. This information will further improve the teaching information of my manuscript and aid other groups for implementing this technique for their own patients.

Line 378 – Thank you for this. I added “pulmonary” here.

Reference 6 – Thank you very much for your attention to detail. I have now provided the reference that I had wanted to place here.